# Expression and Clinical Significance of CD47 in Colorectal Cancer: A Review

**DOI:** 10.3390/cancers18010054

**Published:** 2025-12-24

**Authors:** Qijie Li, Paola Vignali, Donghao Tang, Giulia Martinelli, Beatrice Fuochi, Rebecca Sparavelli, Anello Marcello Poma, Rossella Bruno, Elisabetta Macerola, Clara Ugolini

**Affiliations:** 1Department of Surgical, Medical, Molecular Pathology and Critical Area, University of Pisa, 56126 Pisa, Italy; qijie.li@phd.unipi.it (Q.L.); giulia.martinelli@phd.unipi.it (G.M.); b.fuochi@studenti.unipi.it (B.F.); rebecca.sparavelli@phd.unipi.it (R.S.); marcello.poma@med.unipi.it (A.M.P.); 2Department of Translational Research and New Technologies in Medicine and Surgery, University of Pisa, 56126 Pisa, Italy; paola.vignali@med.unipi.it (P.V.); d.tang1@studenti.unipi.it (D.T.); 3Unit of Pathological Anatomy, University Hospital of Pisa, 56126 Pisa, Italy; r.bruno@ao-pisa.toscana.it (R.B.); elisabetta.macerola@for.unipi.it (E.M.)

**Keywords:** CD47, colorectal cancer, immunohistochemistry, immune escape, prognosis

## Abstract

Colorectal cancer is one of the most common and deadly cancers worldwide. Despite advances in treatment, many patients still experience relapse or poor outcomes, highlighting the need for new immune-based strategies. One promising target is CD47, a molecule on tumor cells that sends a “don’t eat me” signal to immune cells and helps cancer escape immune destruction. However, previous studies have reported inconsistent CD47 expression levels, making its clinical interpretation challenging. This review explores what is currently known about CD47 expression in colorectal cancer, how it interacts with the immune environment, and how it relates to patient prognosis and molecular subtypes. By bringing together findings from different studies, we aim to clarify the potential of CD47 as a biomarker and therapeutic target, providing insights that may support the development of more effective and personalized immunotherapy approaches for colorectal cancer.

## 1. Introduction

CRC ranks as the third most prevalent cancer globally and the second foremost cause of cancer-related mortality, with around 1.8 million new cases and 880,000 deaths annually [1,2]. In addition, a colorectal cancer-like subtype of cancer of unknown primary has recently been described and is managed according to CRC treatment paradigms, which may partly contribute to observed CRC incidence trends [3]. CRC is a disease with significant molecular heterogeneity, among which approximately 15% of the cases show microsatellite instability (MSI) caused by deficient DNA mismatch repair (dMMR) [4,5]. However, both biological and methodological heterogeneity in dMMR assessment using immunohistochemistry (IHC) and MSI testing may lead to discordant results, thereby complicating the interpretation and comparison of immune-related findings across studies [6].

The tumor microenvironment (TME) has received increasing attention for its role in CRC progression and treatment response [7,8]. Immune infiltration patterns affect both tumor heterogeneity and response to immunotherapies such as checkpoint inhibitors [9,10,11,12]. Tumors avoid immune surveillance through multiple mechanisms [13,14,15,16]. Among these, tumor-associated macrophages (TAMs) promote immune escape by inducing T cell (Tregs) exhaustion [17,18]. However, reported CD47 expression levels in CRC vary considerably across studies, highlighting the need for a review of the available evidence. CD47, an innate immune checkpoint frequently overexpressed in CRC, serves as a major “don’t eat me” signal, by binding to SIRPα on the surface of macrophages, thereby inhibiting phagocytosis and facilitating tumor immune evasion [19,20,21,22,23].

Although therapeutic strategies targeting CD47 are being gradually explored, especially in metastatic CRC, its potential as a predictive or prognostic marker remains unclear [24,25]. Moreover, published studies report inconsistent CD47 expression patterns, methodologies, and clinical associations, leaving the overall significance of CD47 in CRC insufficiently defined. Therefore, we conducted this review to systematically summarize the expression characteristics of CD47 in CRC and its relationship with clinicopathological characteristics.

## 2. Mechanism of CD47-Mediated Immune Escape

Tumors achieve immune escape partly through adaptive checkpoints such as Programmed cell death protein 1 (PD-1)/Programmed death-ligand 1 (PD-L1) and Cytotoxic T-lymphocyte-associated protein 4 (CTLA-4), which normally restrain T-cell activity but are co-opted by cancers to induce T-cell exhaustion [26,27,28,29,30,31]. In the TME, immunosuppressive populations (Tregs, MDSCs, TAMs) further enhance PD-1/PD-L1 signaling and promote tumor progression [32,33,34,35].

Studies have found that TAMs not only highly express PD-L1 itself, but also enhance the expression of PD-1 in T cells via cell–cell contact or cytokine release, thereby inducing functional exhaustion of CD8^+^ T cells [36,37,38]. Antigen presentation by TAMs may trigger excessive T-cell activation, which leads to T-cell dysfunction [39,40]. During this process, the Interferon regulatory factor 8 acts as a key regulator [41,42]. Therefore, the activation of immune checkpoint pathways plays a bridging role in immune escape, reflecting not only the immune suppression state of the TME but also revealing the core functions of immune cells such as TAMs in regulating this process [43,44,45].

Beyond adaptive checkpoints, innate immune checkpoints also contribute to tumor immune evasion [46,47]. Among them, the CD47–SIRPα axis has emerged as a central mechanism by which tumor cells suppress macrophage-mediated phagocytosis [48,49]. CD47 is a multi-pass transmembrane protein, with molecular weight between 45 and 55 kDa, composed of an extracellular immunoglobulin superfamily domain, five transmembrane regions, and a short alternatively spliced cytoplasmic tail [50,51,52,53].

As a ligand of inhibitory receptor SIRPα, CD47 is differentially expressed and functions differently between normal and tumor cells, regulates the outcome of myeloid cell-target cells in many homeostasis processes, and plays important roles in cellular functions such as migration, adhesion, angiogenesis, proliferation, and apoptosis [50,54,55,56]. Under physiological conditions, CD47 is broadly expressed and helps maintain self-tolerance, particularly in red blood cells and hematopoietic stem cells [57,58]. In tumor cells, CD47 is frequently overexpressed and binds to the inhibitory receptor SIRPα, primarily expressed on myeloid cells, thus facilitating innate immune escape [59,60]. Moreover, CD47 expression is implicated in epithelial–mesenchymal transition (EMT), tumor stemness, and the shaping of the immunosuppressive microenvironment [61,62,63,64]. Upon CD47 binding, the intracellular immunoreceptor tyrosine-based inhibitory motifs (ITIMs) of SIRPα undergo phosphorylation by Src family kinases [62,64,65], resulting in the recruitment of Src homology region 2 domain-containing phosphatase-1/2 (SHP-1/2), which are normally autoinhibited through Src homology 2 domains [66,67,68]. Binding to phosphorylated ITIMs elicits conformational alterations that activate SHP-1/2 [62,69,70].These phosphatases dephosphorylate downstream targets such as myosin IIA, thereby suppressing cytoskeletal remodeling [53,71]. As a result, macrophage phagocytosis is blocked and tumor cells escape immune clearance [59]. In summary, CD47–SIRPα signaling enables tumor cells to avoid macrophage-mediated clearance, making it a key innate immune checkpoint and an attractive therapeutic target.

Strategies to block the CD47–SIRPα axis include monoclonal antibodies targeting CD47 or SIRPα to disrupt their interaction [72,73,74]; engineered high-affinity SIRPα variants that enhance antibody-dependent phagocytosis [75,76], and gene- or nanotechnology-based approaches to downregulate CD47 expression on tumor cells [25] (Figure 1).

While the majority of research on CD47 blockade utilizes T cell-deficient xenograft models that emphasize macrophage effects, immunocompetent models reveal that CD47 inhibition also promotes CD8^+^ T cell-mediated antitumor responses [77,78,79]. This results from enhanced antigen processing and presentation by dendritic cells (DCs), facilitating superior T cell priming [65,80,81]. Moreover, CD47 inhibits the function of natural killer (NK) cells [81,82]. Collectively, CD47 suppresses both innate and adaptive immunity to promote tumor immune escape [83,84]. In CRC, this may be particularly relevant in MSI-high tumors, where CD47-mediated suppression of antigen presentation could impair the initiation of adaptive anti-tumor responses.

Mechanistically, upregulation of CD47 in tumors is induced by pro-inflammatory cytokines like TNF-α through the activation of NF-κB signaling, signifying an adaptive response to the TME [66,85,86,87]. These findings identify the CD47-SIRPα axis as a crucial target in tumor immunotherapy [72].

## 3. Expression of CD47 in CRC

### 3.1. CD47 Expression in Immunohistochemistry Studies

Three IHC studies have consistently reported that CD47 is markedly elevated in CRC tissues [22,23,88]. Oh et al. evaluated CD47 expression in 468 CRC patient samples, including tumor tissues, normal colonic mucosa, and metastatic/non-metastatic lymph nodes [23]. They found that CD47 was mainly localized on tumor cell membranes (detected in 250 of 468 cases, 53.4%), with low expression in the stroma and normal mucosa. CD47 expression intensity and total IHC scores were considerably elevated in metastatic lymph nodes compared to non-metastatic ones (*p* = 0.029). Fujiwara-Tani et al. assessed 95 stage II–IV CRC samples and reported a high CD47 positive rate of 86.3% (82/95), markedly higher than in normal tissues [22]. Despite inter-patient heterogeneity, no distinct spatial distribution pattern was identified. Tian et al. analyzed 90 colon adenocarcinoma (COAD) cases and found a CD47 expression rate of 91.11%, mainly localized to the cell membrane and cytoplasm [88].

Nonetheless, two studies have shown divergent results [89,90]. Kim et al. analyzed 328 colorectal adenocarcinoma (CRA) samples and identified a low CD47 positive rate of 16.2% (H-score ≥ 50, calculated as 1 × % of weakly, 2 × % of moderately, and 3 × % of strongly stained tumor cells; total range 0–300), with no positive expression in normal mucosa. The average expression score in CRA was considerably elevated compared to normal mucosa (*p* = 0.013) [89]. Sugimura-Nagata et al. reported a CD47 positive rate of 35% on the tumor cell membrane in CRC, which was lower than the rates observed in several other studies [90]. In addition to positivity rate-based analyses, two IHC studies categorized CD47 expression using score-based cut-off values rather than binary positivity. Hu et al. classified CRC cases into CD47-high and CD47-low groups based on an IHC score threshold (<4 vs. ≥4) [91], whereas Aktepe et al. defined CD47 positivity using a median staining index [92]. These methodological differences further highlight the heterogeneity in CD47 assessment across IHC studies. Characteristics of included studies are summarized in Table 1.

### 3.2. CD47 Expression in Transcriptomic Analyses

Recent studies utilizing public databases, including The Cancer Genome Atlas (TCGA), Gene Expression Omnibus (GEO), and Gene Expression Profiling Interactive Analysis (GEPIA), have examined CD47 expression profiles and their clinical implications in CRC. These studies consistently demonstrate increased CD47 expression in CRC tissues and its correlation with numerous clinicopathological characteristics, TME, and prognosis.

Tian et al. found that high CD47 expression was substantially correlated with altered M1/M2 macrophage ratios subtypes in CRC, using TCGA-COAD data [88]. Furthermore, co-expression of CD47 with adaptive immune checkpoint molecules such as PD-1 and PD-L1 was significantly elevated, suggesting a pivotal function for CD47 in tumor immune evasion. Oh et al. revealed that CD47 mRNA levels were considerably elevated in CRC tissues compared to matched normal mucosa (*p* = 0.048) via Reverse Transcription Polymerase Chain Reaction (RT-PCR) [23]. These results suggest CD47 is transcriptionally upregulated.

Consistent elevation of CD47 at both mRNA and protein levels has been documented in various studies, indicating a robust molecular correlation. Importantly, this transcriptional upregulation is consistently observed across independent datasets employing different platforms, reinforcing its reliability beyond technical artifacts. Prior research has associated elevated CD47 expression with EMT, stem-like transcriptional characteristics, and PD-1/PD-L1 resistance [94]. These associations indicate that CD47 may aid in diagnosis, molecular subtyping, and personalized CRC therapy.

Arai et al. performed a large-scale molecular profiling study of 14,287 CRC cases and discovered that CD47-high tumors were markedly enriched for CMS1 and CMS4 subtypes, both linked to immune activation and heightened immune cell infiltration [93]. Elevated CD47 expression correlated with the activation of oncogenic pathways (e.g., Mitogen-activated protein kinase, Phosphoinositide 3-kinase, TGF-β, angiogenesis) and upregulation of multiple immune checkpoints, including PD-1, PD-L1, CTLA-4, and Lymphocyte-activation gene 3 (LAG3). Consistently, Tumor Immune Estimation Resource (TIMER)-based analysis of TCGA-COAD data confirmed increased infiltration of CD8^+^ T cells, macrophages, and dendritic cells in CD47-high tumors, supporting a potential role for CD47 in modulating an immunosuppressive TME [95].

## 4. Clinical and Prognostic Correlations of CD47 Expression in CRC

### 4.1. Correlation Between CD47 and TNM Staging

Tian et al. examined 90 COAD cases and found significantly higher CD47 expression in patients with T3–T4, N1–N2, and Tumor–Node–Metastasis (TNM) stage III–IV tumors (all *p* < 0.05), supporting a positive correlation with disease advancement [88]. Similarly, Oh et al. reported in 468 CRC samples that elevated CD47 expression was significantly associated with deeper tumor invasion (*p* = 0.009), lymph node metastasis (*p* < 0.001), and distant metastasis (*p* = 0.004) [23]. These findings suggest that CD47 is elevated in advanced TNM stages. In another cohort of 328 CRC patients, Kim et al. used an H-score system and showed an association between CD47 expression and AJCC staging, although specific *p* values were not provided [89].

### 4.2. Correlation Between CD47 and Tumor Differentiation

Oh et al. found a significant correlation between CD47 expression and tumor differentiation, with higher expression noted in poorly differentiated tumors (*p* = 0.020) [23]. Sugimura-Nagata et al. similarly found a higher prevalence of poorly differentiated adenocarcinoma and mucinous adenocarcinoma subtypes in CD47-positive patients (*p* = 0.032 and *p* = 0.040, respectively) [90]. Nonetheless, Fujiwara-Tani et al. reported no statistically significant correlation between CD47 expression and tumor differentiation (*p* = 0.289), indicating that differences in cohort composition and scoring methodologies may affect the observed associations [22]. Other IHC studies primarily focused on prognostic stratification rather than histological grading and did not consistently report independent associations between CD47 expression and tumor differentiation [91,92].

### 4.3. Correlation Between CD47 and Lymphatic/Distant Metastasis

Oh et al. pointed out that high CD47 expression was notably correlated with both lymph node metastasis (*p* < 0.001) and distant metastasis (*p* = 0.004) [23]. Tian et al. also noted increased CD47 expression in N1/N2 stage patients [88]. Although Fujiwara-Tani et al. found a significant correlation between CD47 and metastasis, they did not find a comparable relationship with local T staging (*p* = 0.472) [22]. Additionally, Kim et al. discovered that CD47 expression was notably associated with multiple metastasis-related markers, such as lymphovascular invasion, perineural invasion, and tumor budding [89]. Similarly, Aktepe et al. observed that CD47-positive tumors exhibited more aggressive clinicopathological features and poorer outcomes, indirectly supporting its association with metastatic potential [92]. Moreover, Sugimura-Nagata et al. reported heightened infiltration of SIRPα and CD163 TAIs in CD47-positive samples, indicating that CD47 positivity is linked to macrophage infiltration and immune evasion [90].

### 4.4. Prognostic Impact of CD47 Expression on Survival and Recurrence

Oh et al. identified a correlation between elevated CD47 expression and reduced OS in univariate analysis; however, this association lacked significance after controlling for age, sex, tumor size, and TNM staging in multivariate Cox regression [23]. Tian et al. reported a paradoxical finding: while high CD47 expression was associated with advanced TNM stage, it also correlated with longer progression-free survival (PFI) (HR = 0.57, *p* = 0.025) in their cohort [88]. The authors themselves noted this discrepancy and investigated it, finding a positive correlation between CD47 expression and markers of M1-polarized macrophages (e.g., CD86, CXCL9). They therefore speculated that in this specific context, the prognostic signal of CD47 might be modulated or overridden by a concomitant anti-tumor (M1) immune response, highlighting the microenvironment-dependent nature of its role. Fujiwara-Tani et al. found no significant correlation between CD47 expression and overall survival (*p* = 0.289) in colorectal cancer patients [22]. In another study, Kim et al. reported that while high CD47 expression was associated with shorter recurrence-free survival (RFS) in univariate analysis (HR = 1.792, 95% CI 1.014–3.084; *p* = 0.035), it did not retain independent prognostic significance in multivariate analysis when adjusted for AJCC stage and treatment (HR = 1.406, 95% CI 0.815–2.428; *p* = 0.221). No significant association was found with cancer-specific survival (CSS) [89]. In contrast, Sugimura-Nagata et al. revealed that the 5-year survival was significantly lower in CD47-positive patients (64.0% vs. 79.0%, *p* = 0.0268) [90]. Multivariate Cox regression identified high CD47 expression as an independent adverse prognostic factor (HR = 1.75, *p* = 0.038), comparable to lymph node (HR = 2.26) and peritoneal metastasis (HR = 5.80). Consistent with these findings, Hu et al. analyzed a tissue microarray-based CRC cohort and demonstrated that CD47-high expression was significantly associated with both shorter overall survival and disease-free survival, remaining an independent adverse prognostic factor in multivariate Cox regression analyses [91]. Similarly, Aktepe et al. reported that CD47-positive tumors were independently associated with poorer overall survival after adjustment for clinicopathological variables, further supporting the prognostic relevance of CD47 in CRC [92]. In a large cohort of 14,287 CRC cases from the Caris Life Sciences database, Arai et al. reported that high CD47 tumors were enriched in CMS1 and CMS4, both associated with immune activation and unfavorable outcomes [93]. The prognostic findings from published studies are summarized in Table 1.

Collectively, CD47 expression shows consistent prognostic associations in CRC, although its effect may be modulated by the immune microenvironment, molecular subtype (e.g., CMS1/4), and co-expression of other immune checkpoints. Future studies ought to focus on integrative multi-omics methodologies to clarify the independent predictive value of CD47 and assess its potential utility as a personalized prognostic biomarker.

### 4.5. Correlation Between CD47 Expression and Consensus Molecular Subtypes

The CMS system stratifies CRC into four subtypes with distinct transcriptional and immune features. Among the four established subtypes, CMS1 and CMS4 are closely linked to immune microenvironmental characteristics and exhibit distinct expression levels of CD47 [93].

Arai et al. performed an analysis utilizing 14,287 CRC samples. It was found that the high expression of CD47 mRNA was significantly enriched in the CMS1 (characterized by hypermutation, high microsatellite instability, pronounced immunogenicity, mutations of the BRAF gene) and CMS4 (activation of the TGF-β signaling pathway, epithelial–mesenchymal transition, severe stromal infiltration, active neoangiogenesis, and poor prognosis) [93,96]. These two subtypes belong to the “immune hot” type [97], but they exhibit a paradoxical relationship with immune escape, indicating that CD47 may function as a compensatory “Don’t eat me” signal within the inflammatory microenvironment.

Valenzuela et al. revealed that patients with CMS4 exhibited heightened expression of genes associated with matrix and angiogenesis, diminished response to chemotherapy, and reduced OS [95]. The high expression of CD47 in the CMS4 subtype may enhance its immunosuppressive and treatment resistance. As an innate immune checkpoint, high CD47 expression may contribute to immune exclusion in CMS4 tumors [98]. This subtype is also marked by strong matrix remodeling and angiogenesis-related gene expression [99,100]. Although the early prognosis of CMS1 patients is relatively good, their survival significantly decreases after recurrence, which may be related to the persistent expression of immune checkpoints, including CD47. By delivering a potent “Don’t eat me” signal, CD47 suppresses macrophage phagocytosis and dendritic cell antigen presentation. This innate immune checkpoint activity likely synergizes with adaptive checkpoints (e.g., PD-L1) to create a compounded immunosuppressive barrier, facilitating immune evasion and disease progression upon recurrence. Data from Arai et al. show that CD47-high tumors are enriched in CMS1 and CMS4 subtypes, and key CMS subtype features are summarized according to Valenzuela et al., which are summarized in Table 2.

Overall, the prognosis and therapeutic significance of CD47 may depend on the CMS context. Future research should incorporate CMS classification when analyzing CD47 expression to better predict prognosis and guide immunotherapy, especially for high MSI and interstitial CRC.

### 4.6. The Relationship Between CD47 and the Tumor Immune Microenvironment

Multiple studies have shown that the high expression of CD47 in CRC is closely related to specific immune cell infiltration patterns and co-expression of immune checkpoints, suggesting that it plays an important role in shaping the TME.

Tian et al. found through TIMER analysis based on TCGA-COAD data that the infiltration of CD8^+^ T cells, macrophages, and dendritic cells were significantly increased in tumors with high CD47 expression [88]. CD47 expression correlated positively with M1 macrophage markers (CD86, CXCL9, CD80; *p* < 0.001) but negatively with M2 markers (*p* < 0.001), suggesting context-dependent effects. Furthermore, Sugimura-Nagata et al. reported that in CD47-positive CRC samples, the numbers of SIRPα and CD163 cells in TAIs significantly increased (*p* = 0.044; *p* < 0.0001), which was consistent with the enrichment of immunosuppressive macrophages [90]. The increase in CD163+ cells, a hallmark of M2-like, pro-tumorigenic TAMs, suggests that CD47 may actively shape an immunosuppressive TME by promoting the polarization or recruitment of such cells. These findings indicate that CD47 not only facilitates innate immune evasion via the CD47-SIRPα axis but may also enhance the recruitment of immunosuppressive myeloid cells, thus establishing an immune-tolerant TME. This paradox might reflect dysfunctional or “exhausted” macrophages that express M1 markers but have lost anti-tumor function, possibly due to chronic CD47-SIRPα signaling.

## 5. Discussion

Although an increasing amount of evidence indicates that CD47 is a promising biomarker and immunotherapy target, there are still some issues that need to be addressed in its clinical translation. More broadly, the clinical implementation of emerging biomarkers in colorectal cancer is often limited by a lack of standardization in molecular testing, tumor heterogeneity, and real-world constraints such as cost and access, which may hinder consistent interpretation and clinical applicability [101].

First, there are significant differences in the positive rate of CD47 recorded in different IHC studies, ranging from 16.2% to 91.1%. This heterogeneity may stem from multiple methodological factors, including: (a) Antibody differences: These five studies used different antibody clones or antibodies from different companies (such as Abcam, CST, different or unspecified), which differed in epitope affinity and signal strength; (b) Scoring system: The definition of high expression varies in different studies. Some studies use an H-score of ≥50, while others use a combination of percentage thresholds, intensity and area scores or do not specify them. For example, a sample with 60% weak (1+) staining would score 60 by H-score (1 × 60), qualifying as positive per Kim et al.’s cutoff (H-score ≥ 50). However, under a scoring system that emphasizes staining intensity, the same weak staining might only score 1 point, potentially classifying it as low or negative (e.g., as implied in intensity-based methods like Tian et al.’s [88]). This directly illustrates how scoring criteria alone can dramatically alter positivity rates. (c) Cut-off value: Arbitrary or research-specific positive thresholds hinder comparisons between studies. (d) Cohort size: The included studies varied considerably in cohort size (ranging from *n* = 90 to *n* = 468). Smaller cohorts are inherently more susceptible to random sampling error and have lower statistical power, which may partially explain divergent results in studies of CD47’s correlation with specific clinicopathological features or its prognostic value. These inconsistencies reflect the urgent need to develop a standardized assessment plan for CD47 IHC. Future meta-analyses or cross-platform TCGA comparisons may be helpful in determining reliable evaluation parameters and standardizing cross-cohort expressions. In addition to methodological factors, differences in inclusion criteria and tumor subtypes can also lead to heterogeneity. Some studies only evaluate COAD, while others assess all CRC (colon and rectum), or specifically evaluate CRA. These differences cannot be ignored: Right-sided colon cancers are more frequently characterized by MSI-high and CMS1 subtypes, whereas rectal cancers are predominantly CMS2/3, which can reshape the TME and may indirectly affect the expression of CD47 [102,103]. This is biologically grounded in their distinct embryological origins (midgut vs. hindgut), which establish divergent immune milieus that likely modulate checkpoint expression. In addition, pathological variant types (CRA, mucinous, serrated) exhibit different immune infiltration characteristics, further altering the association between CD47 and prognosis [104,105,106]. In conclusion, these biological and clinical differences have led to variations in the results among studies.

Second, most of the existing studies are retrospective, which essentially limits the reliability and universality of their research results. Retrospective analysis is prone to be influenced by factors such as patient selection, heterogeneity of detection methods, and data deficiency. Therefore, the association between the reported CD47 expression and adverse pathological features (including advanced TNM stage, lymphovascular invasion and tumor budding) should be interpreted with caution [107], as the current evidence only indicates a correlation rather than a causal relationship. It is still uncertain whether the high expression of CD47 really promotes tumor invasiveness or is merely a coincidence. Therefore, the lack of prospective clinical validation and mechanism studies further limits the ability to establish a causal relationship between high CD47 in CRC and adverse clinicopathological outcomes. Future prospective studies using standardized CD47 assessment and predefined clinical endpoints will be essential to clarify causal relationships between CD47 expression and clinical outcomes.

Third, conventional methodologies such as bulk RNA-sequencing and IHC have limitations because they only measure average expression levels without distinguishing between cell types or spatial distribution in the TME [108,109]. This simplification method cannot reflect the complex cellular and spatial heterogeneity specific to CRC [110]. Recent studies have shown that the biological effects of CD47 may vary depending on the immune environment [89,111]. Especially, the balance between M1 and M2 type TAMs may affect tumor progression in the opposite way [88]. Therefore, transcriptome analysis indicated that CD47 was abundantly expressed in the CMS1 (MSI-immune) and CMS4 (mesenchymal) subtypes, both of which are characterized by intense immune infiltration [93]. But paradoxically, they are associated with poor prognosis. These findings suggest that CD47-mediated immune escape may be environmentally dependent and influenced by macrophage polarization and molecular subtypes [112]. Specifically, in tumors dominated by immunosuppressive M2-like TAMs, high CD47 expression would reinforce an existing “Don’t eat me” barrier, aggravating immune evasion and correlating with poor outcomes. Conversely, in tumors with a high infiltration of inflammatory M1-like macrophages, elevated CD47 might represent a compensatory, but ultimately insufficient, response by tumor cells to resist phagocytic pressure. In this context, the prognostic signal of CD47 could be confounded or even reversed by the dominant anti-tumor immune activity, as suggested by Tian et al. [88]. Thus, the net prognostic value of CD47 is not intrinsic but is interpreted through the lens of the local myeloid cell composition. Advanced methods such as single-cell transcriptomics and spatial proteomics are crucial for clarifying these interactions and improving the prognostic and therapeutic value of CD47.

Recent spatial transcriptomic studies have improved the understanding of the colorectal cancer tumor microenvironment by revealing spatially distinct immune and stromal compartments, which cannot be captured by bulk transcriptomic analyses [113].

In addition to its immunoregulatory role, CD47 has been reported to exert tumor-intrinsic, immune-independent functions in colorectal cancer [91]. Hu et al. demonstrated that CD47 overexpression promotes CRC cell proliferation, invasion, and metastasis in vitro and in vivo, even in macrophage-depleted nude mouse models, indicating that these effects are independent of the CD47–SIRPα immune axis [91]. Mechanistically, CD47 was shown to interact with the glycolytic enzyme ENO1, protecting it from FBXW7-mediated ubiquitin degradation [114]. Stabilization of ENO1 was associated with enhanced aerobic glycolysis and ERK signaling, thereby facilitating tumor growth and metastatic potential [91,115]. Collectively, these findings expand the functional landscape of CD47 beyond immune evasion and suggest that CD47 may also contribute to CRC progression through metabolic reprogramming and tumor-intrinsic signaling pathways.

Preclinical studies using experimental models have provided important evidence supporting the therapeutic potential of CD47 targeting in colorectal cancer. In vitro studies in CRC cell lines and in vivo xenograft models have shown that genetic or pharmacological modulation of CD47 can suppress tumor growth and metastatic potential, even under immune cell-depleted conditions, highlighting tumor-intrinsic vulnerabilities [91]. In addition, translational analyses have suggested that CD47-directed strategies may enhance therapeutic efficacy and support further development of CD47-targeting approaches in CRC [92].

Fourth, the rationale for simultaneously blocking the CD47–SIRPα and PD-1/PD-L1 pathways is strongly supported by emerging translational evidence. This dual-targeting strategy has been validated by a CD47/PD-L1 bispecific antibody (6MW3211), which demonstrated potent synergistic anti-tumor efficacy in preclinical models and a favorable safety profile in GLP non-human primate studies, underscoring its clinical translation potential [116]. Furthermore, histological analysis confirmed the co-expression of PD-L1 and CD47 on several human tumor tissues, providing a biomarker rationale for patient selection in clinical trials [117]. In this context, our discovery of SMC18 (a novel small-molecule dual inhibitor of both CD47/SIRPα and PD-1/PD-L1) provides a complementary therapeutic approach with a distinct pharmacokinetic profile to overcome potential cross-resistance associated with single-pathway blockade. The translational potential of co-targeting these pathways is underscored by ongoing clinical efforts, such as a phase I/II trial combining a CD47 antibody with a small-molecule TLR7 agonist and chemotherapy in advanced solid tumors (NCT04588324). This validates the clinical rationale for our dual-targeting strategy. In this context, SMC18—a single small-molecule inhibitor of both CD47/SIRPα and PD-1/PD-L1—represents a direct and integrated approach to overcome potential cross-resistance. Future research should explore the interaction mechanism between the two pathways to investigate potential synergistic or redundant effects, thereby potentially discovering new targets for combined immunotherapy.

Emerging clinical data indicate that CD47-targeted strategies are being actively evaluated in CRC, with early results suggesting heterogeneous activity and notable safety considerations. In an open-label phase 1b/2 study, the anti-CD47 antibody magrolimab combined with cetuximab showed modest efficacy in heavily pretreated CRC, with ORR 6.3% in KRAS wild-type tumors versus 0% in KRAS-mutant tumors (DCR 50.0% vs. 38.1%), and manageable toxicities; grade ≥ 3 anemia occurred in 11.5% of patients and no treatment-related deaths were reported [118]. In contrast, a phase II trial testing evorpacept (ALX148) plus cetuximab and pembrolizumab in refractory MSS mCRC reported an ORR of 6.3% and a DCR of 12.5%, but enrollment was terminated early due to serious immune-related toxicities, including treatment-related grade 5 hemophagocytic lymphohistiocytosis and cytokine release syndrome [119]. Collectively, these findings support the feasibility of CD47-based combinations in CRC while underscoring the need for careful patient selection and safety optimization.

Ultimately, although CD47 is associated with poor prognosis in some cohorts [120], its predictive value for treatment response, especially to immune checkpoint inhibitors, still awaits verification through clinical trials. Combining CD47 expression with multi-dimensional immune features (such as CMS, TAM polarization, and interferon features) may help develop more accurate risk stratification models.

## 6. Conclusions

CD47 is widely overexpressed in CRC and is associated with adverse pathological features, immune infiltration and prognosis. However, the inconsistencies in detection methods, scoring criteria and cohort characteristics have limited its clinical application. We should further focus on standardizing the unified IHC assessment protocol, including the selection of antibody clones, scoring criteria and positive cut-off values.

Integrating CD47 expression data with single-cell and spatial transcriptomics methods may help us better understand the role of CD47 in immune escape. Furthermore, co-targeting CD47 with adaptive checkpoints such as PD-1 or LAG3 holds promising therapeutic prospects, especially CMS4 CRC. Overcoming issues such as extratumoral toxicity and weak phagocytic activation remains a key challenge in the development of anti-CD47 antibodies.

## Figures and Tables

**Figure 1 cancers-18-00054-f001:**
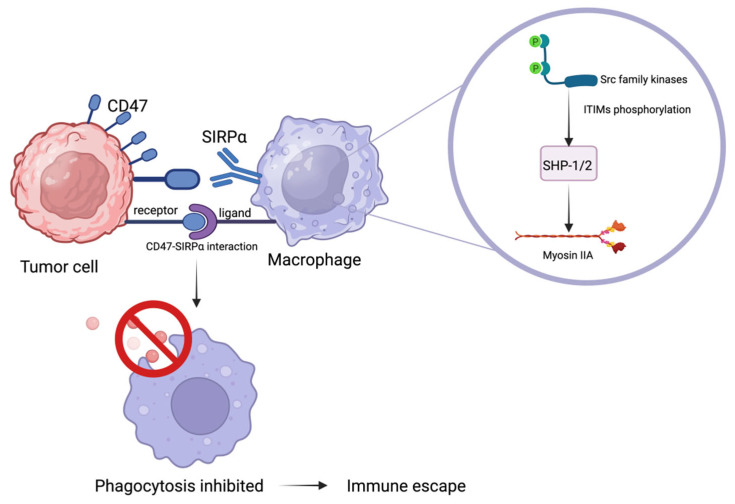
CD47 expressed on tumor cells binds SIRPα on macrophages, activating ITIM–SHP-1/2 signaling that blocks cytoskeletal remodeling and phagocytosis, thereby transmitting a “don’t eat me” signal. Created with BioRender.com. https://BioRender.com/joi5t5i.

**Table 1 cancers-18-00054-t001:** Summary of studies evaluating the prognostic significance of CD47 in colorectal cancer.

Study	Cohort	Method	Sample Size	CD47 Assessment	Staining Localization	Scoring Method	Antibody Used	Main Finding (Standardized)
Oh et al. [23]	CRC (South Korea)	IHC	468	53.4% (250/468)	Cell membrane	Intensity (0–3) × Density (1–4); Mean IHC score ≥ 5 = positive	Clone not specified, Abcam, Cambridge, UK;	OS ↓ (poorer OS in univariate; not significant in multivariate)
Fujiwara-Tani et al. [22]	CRC (Japan)	IHC	95	86.3% (82/95)	Cell membrane	Staining index = intensity (0–3) × area (%); intensity 1 = normal tissue baseline	Clone not specified, Abcam, Cambridge, UK;	OS –
Tian et al. [88]	COAD (China)	IHC	90	91.11% (80/90)	Cell membrane and cytoplasm	A scale of 0 to 3: 0 points for no staining, 1 point for 1–25% positive staining, 2 points for 26–50%, and 3 points for 51–100%. High CD47 expression was defined by an IHC score of 3 or higher.	1:200, CST, 63000S	OS –; DSS –; PFI ↑
Kim et al. [89]	CRA (South Korea)	IHC	328	16.2% (53/328)	Membrane	H-score = 1 × (% of 1 + cells) + 2 × (% of 2 + cells) + 3 × (% of 3 + cells). Positive expression groups (H-score ≥ 50)	Abcam, Cambridge, UK, EPR21794 (1:200)	CSS ↓; RFS ↓ (high CD47 expression associated with poorer CSS and shorter RFS)
Sugimura-Nagata et al. [90]	CRC (Japan)	IHC	269	35% (95/269)	Cytomembrane	Not specified	SP279, Abcam (Cambridge, UK)	5-year survival ↓ in CD47-positive tumors; high CD47 was an independent risk factor
Arai et al. [93]	CRC (USA)	Whole-transcriptome RNA sequencing analysis	14,287	NA	NA	NA	NA	OS—(CD47 mRNA not correlated with OS; enriched in CMS1/4)
Hu et al. [91]	CRC (China)	IHC (TMA)	293	High vs. Low (IHC score cut-off: <4 vs. ≥4)	NA	IHC score (cut-off: <4 vs. ≥4)	NA	OS ↓; DFS ↓ (CD47-high associated with poorer OS and DFS; independent prognostic factor in multivariate analysis)
Aktepe et al. [92]	CRC (Turkey)	IHC (TMA)	98	56.1% (55/98)	Membranous staining	Area score (1–4 by % cells: 1% = 1–25%, 2 = 26–50%, 3 = 51–75%, 4 > 75%) × Intensity (0–3+); staining index = intensity × area; median IHC score = 4; negative < median, positive ≥ median	Anti-CD47, Abcam (Cambridge, UK)	OS ↓ (CD47-positive had shorter OS; independent prognostic factor in multivariate Cox: HR 2.142, *p* = 0.006)

Abbreviations: CD47, cluster of differentiation 47; CRC, colorectal cancer; COAD, colon adenocarcinoma; CRA, colorectal adenocarcinoma; IHC, immunohistochemistry; TMA, tissue microarray; OS, overall survival; DSS, disease-specific survival; PFI, progression-free interval; CSS, cancer-specific survival; RFS, recurrence-free survival; CMS, consensus molecular subtype; NA, not applicable. Legend: ↓ indicates worse outcome for CD47-high or CD47-positive tumors; ↑ indicates better outcome; – indicates no significant association.

**Table 2 cancers-18-00054-t002:** Correlation between CD47 expression and CMS in CRC.

CMS Subtype	Predominance	CD47 Association	Key Features
CMS1 (Immune)	Proximal colon	Enriched in CD47-high tumors (17.9% vs. 14.5%)	MSI-high, high CIMP/BRAF mutations, low prevalence of SNCA, immune infiltration
CMS2 (Canonical)	Distal colon and rectum	Reduced in CD47-high tumors (31.1% vs. 35.4%)	Epithelial phenotype, WNT/MYC activation, chromosomal instability
CMS3 (Metabolic)	Without predominance	Reduced in CD47-high tumors (10.9% vs. 23.3%)	KRAS mutations, metabolic reprogramming, intermediate MSI/CIMP
CMS4 (Mesenchymal)	Distal colon and rectum	Enriched in CD47-high tumors (40.1% vs. 26.8%)	Stromal infiltration, EMT/TGF-β activation, angiogenesis, poor prognosis

## Data Availability

No new data were created or analyzed in this study. Data sharing is not applicable to this article.

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
