# Peer review of "Expression and Clinical Significance of CD47 in Colorectal Cancer: A Review"

_cancers, 2025, doi:10.3390/cancers18010054_

Round 1

Reviewer 1 Report

Comments and Suggestions for Authors

The esteemed group of authors led by Qijie Li has written a very important and engaging review. Colorectal cancer is a global concern, ranking among the leading causes of cancer incidence. Additional challenges relate to the complexity of diagnosis and treatment, while the prognosis for a significant number of patients remains poor due to late diagnosis and recurrence. Contemporary research focuses on understanding the molecular and genetic mechanisms underlying colorectal cancer development, enabling the creation of new diagnostic markers and more effective targeted therapies. Key directions include studying the tumor immune microenvironment, mechanisms of tumor immune evasion, and the development of immunotherapeutic approaches. Notably, the marker CD47 attracts special attention. Therefore, I find the authors’ review very interesting and timely.

The review is clearly structured, with the material presented in a well-organized logical order. It contains an brief introduction and discussion, main sections dedicated to the mechanism of CD47-mediated immune escape, expression of CD47 in colorectal cancer, as well as clinical and prognostic correlations of CD47 expression in colorectal cancer. The review fully meets its stated objectives and is sufficiently comprehensive, including all major scientific works focused on CD47 research in colorectal cancer. No similar reviews have been published during the past decade or earlier. The review combines fundamental aspects with information valuable for practicing physicians, especially regarding the prognostic role of CD47.

The conclusions drawn by the authors are sound and supported by references. The review includes 106 sources dating from 2009 onwards, with 60% published in the past five years, thus containing the most up-to-date information.

The review also features an original figure schematically explaining the role of CD47 in tumor cell-macrophage interactions, along with three tables summarizing key scientific works.

I highly appreciate the authors’ work and believe it should definitely be published. I have only one minor comment concerning abbreviations: I suggest avoiding abbreviations in the titles of subsections 3.1, 4.5, and 4.6 to enhance readability. Additionally, please provide footnotes explaining all abbreviations found in the tables (e.g., Table 1: CST; Table 2: OS, DSS, PFI, RFS, COAD eyc; Table 3: MSI, CMS). Also, please verify the definitions of abbreviations appearing for the first time in the text, such as IHC (line 123), CRA (line 136), CST (Table 1).

Author Response

REVIEWER 1:

R: The esteemed group of authors led by Qijie Li has written a very important and engaging review. Colorectal cancer is a global concern, ranking among the leading causes of cancer incidence. Additional challenges relate to the complexity of diagnosis and treatment, while the prognosis for a significant number of patients remains poor due to late diagnosis and recurrence. Contemporary research focuses on understanding the molecular and genetic mechanisms underlying colorectal cancer development, enabling the creation of new diagnostic markers and more effective targeted therapies. Key directions include studying the tumor immune microenvironment, mechanisms of tumor immune evasion, and the development of immunotherapeutic approaches. Notably, the marker CD47 attracts special attention. Therefore, I find the authors’ review very interesting and timely.

The review is clearly structured, with the material presented in a well-organized logical order. It contains an brief introduction and discussion, main sections dedicated to the mechanism of CD47-mediated immune escape, expression of CD47 in colorectal cancer, as well as clinical and prognostic correlations of CD47 expression in colorectal cancer. The review fully meets its stated objectives and is sufficiently comprehensive, including all major scientific works focused on CD47 research in colorectal cancer. No similar reviews have been published during the past decade or earlier. The review combines fundamental aspects with information valuable for practicing physicians, especially regarding the prognostic role of CD47.

The conclusions drawn by the authors are sound and supported by references. The review includes 106 sources dating from 2009 onwards, with 60% published in the past five years, thus containing the most up-to-date information.

The review also features an original figure schematically explaining the role of CD47 in tumor cell-macrophage interactions, along with three tables summarizing key scientific works.

A: We sincerely thank the reviewer for the positive and encouraging evaluation of our manuscript and for the helpful suggestions regarding clarity and presentation.

R: I highly appreciate the authors’ work and believe it should definitely be published. I have only one minor comment concerning abbreviations: I suggest avoiding abbreviations in the subsections 3.1, 4.5, and 4.6 to enhance readability.

A: Thank you for the suggestions. We have removed the abbreviations as requested.

R: Additionally, please provide footnotes explaining all abbreviations found in the tables (e.g., Table 1: CST; Table 2: OS, DSS, PFI, RFS, COAD eyc; Table 3: MSI, CMS). Also, please verify the definitions of abbreviations appearing for the first time in the text, such as IHC (line 123), CRA (line 136), CST (Table 1).

A: Thank you. We have carefully revised all tables to ensure clarity and readability. All abbreviations used in the tables (e.g., LN, OS, DSS, PFI, CSS, RFS, CMS, IHC, and TAIs) are now clearly explained in the corresponding table footnotes. We added immunohistochemistry for the first time in the text (Line 53) and CRA is defined at its first occurrence in the main text (Line 152). Company names (e.g., CST and Abcam) are retained as proper nouns and are not treated as abbreviations. All revisions are highlighted in the revised manuscript.

Reviewer 2 Report

Comments and Suggestions for Authors

In this article, the authors summarize the expression characteristics of CD47 in CRC and its relationship with clinicopathological characteristics. The manuscript is straightforward, well written, and concise, within the scope of a review article. Definitely deserves to be published and is a valuable contribution to the “cancers” journal.

However, the following comments need to be addressed, as recommended.

[1] “1. Introduction”, Page 2 of 17, Lines 42-44:

“CRC ranks as the third most prevalent cancer globally and the second foremost cause of cancer-related mortality, with around 1.8 million new cases and 880,000 deaths annually [1,2].”.

From an epidemiological standpoint, it should be emphasized that novel favorable subsets of cancers of unknown primary (CUP) have recently been delineated, most notably a CRC–like CUP subtype. This distinct clinical entity, which is managed according to CRC treatment paradigms, likely accounts in part for the rising incidence trends observed in CRC.

Recommended reference: Mathew BG, et al. From Biology to Diagnosis and Treatment: The Ariadne's Thread in Cancer of Unknown Primary. Int J Mol Sci. 2023;24(6):5588.

[2] “1. Introduction”, Page 2 of 17, Lines 44-46:

“CRC is a disease with significant molecular heterogeneity, among which approximately 15% of the cases show microsatellite instability (MSI) caused by deficient DNA mismatch repair (dMMR) [3,4].”.

At that point, the authors should mention that immune cell PD-L1 expression is significantly higher in dMMR (MSI-H) CRC as compared to MMR-proficient (MSI-L) tumors, with no differences among the different MSI-H molecular subtypes. The recommended screening for defective, DNA MMR includes immunohistochemistry (IHC) and/or MSI test. However, there are challenges in distilling the biological and technical heterogeneity of MSI testing down to usable data. It has been reported in the literature that IHC testing of the MMR machinery may give different results for a given germline mutation and has been suggested that this may be due to somatic mutations.

Recommended reference: Adeleke S, et al. Microsatellite instability testing in colorectal patients with Lynch syndrome: lessons learned from a case report and how to avoid such pitfalls. Per Med. 2022;19(4):277-286.

[3] “4.4. Prognostic Impact of CD47 Expression on Survival and Recurrence”, Page 6 of 17, Lines 206-208:

“Oh et al. identified a correlation between elevated CD47 expression and reduced OS in univariate analysis; however, this association lacked significance after controlling for age, sex, tumor size, and TNM staging in multivariate Cox regression [21].”.

The authors should include a comment regarding the population of elderly patients. Please note that there is no consensus that age independently affects survival outcomes. The prognosis in older individuals may be influenced by confounding factors such as stage at presentation, tumor site, preexisting comorbidities, and the type of treatment received.

Recommended reference: Osseis M, et al. Surgery for T4 Colorectal Cancer in Older Patients: Determinants of Outcomes. J Pers Med. 2022;12(9):1534.

[4] “5. Discussion”, Page 9 of 17, Lines 278-280:

“Although an increasing amount of evidence indicates that CD47 is a promising biomarker and immunotherapy target, there are still some issues that need to be addressed in its clinical translation.”.

At that stage, the authors should note that while tumor-agnostic biomarkers hold great promise for personalized CRC therapy, their clinical implementation faces key challenges. Lack of standardization in molecular testing—stemming from differences in gene panels, sequencing methods, and bioinformatics pipelines—leads to inconsistencies in biomarker detection and interpretation. Additionally, tumor heterogeneity and adaptive resistance, such as bypass signaling in BRAF-mutant CRC, complicate the development of durable treatment strategies. Finally, high testing costs and unequal access to genomic profiling limit the real-world applicability and generalizability of agnostic biomarkers across diverse patient populations.

Recommended reference: Kyrochristou I, et al. Agnostic Biomarkers and Molecular Signatures in Colorectal Cancer-Guiding Chemotherapy and Predicting Response. Biomedicines. 2025;13(8):2038.

Author Response

REVIEWER 2:

R: In this article, the authors summarize the expression characteristics of CD47 in CRC and its relationship with clinicopathological characteristics. The manuscript is straightforward, well written, and concise, within the scope of a review article. Definitely deserves to be published and is a valuable contribution to the “cancers” journal.

A: We thank the reviewer for their constructive comments. The manuscript has been revised accordingly, and all changes are highlighted in the revised version.

R: However, the following comments need to be addressed, as recommended.

“1. Introduction”, Page 2 of 17, Lines 42-44:

“CRC ranks as the third most prevalent cancer globally and the second foremost cause of cancer-related mortality, with around 1.8 million new cases and 880,000 deaths annually [1,2].”.

From an epidemiological standpoint, it should be emphasized that novel favorable subsets of cancers of unknown primary (CUP) have recently been delineated, most notably a CRC–like CUP subtype. This distinct clinical entity, which is managed according to CRC treatment paradigms, likely accounts in part for the rising incidence trends observed in CRC.

Recommended reference: Mathew BG, et al. From Biology to Diagnosis and Treatment: The Ariadne's Thread in Cancer of Unknown Primary. Int J Mol Sci. 2023;24(6):5588.

A: Thank you for this insightful suggestion. We have added a brief statement in the Introduction acknowledging the recently described colorectal cancer–like subtype of cancer of unknown primary and its potential contribution to CRC incidence trends, with the recommended reference included. This revision appears on Lines 47–49 of the revised manuscript.

R: “1. Introduction”, Page 2 of 17, Lines 44-46:

“CRC is a disease with significant molecular heterogeneity, among which approximately 15% of the cases show microsatellite instability (MSI) caused by deficient DNA mismatch repair (dMMR) [3,4].”.

At that point, the authors should mention that immune cell PD-L1 expression is significantly higher in dMMR (MSI-H) CRC as compared to MMR-proficient (MSI-L) tumors, with no differences among the different MSI-H molecular subtypes. The recommended screening for defective, DNA MMR includes immunohistochemistry (IHC) and/or MSI test. However, there are challenges in distilling the biological and technical heterogeneity of MSI testing down to usable data. It has been reported in the literature that IHC testing of the MMR machinery may give different results for a given germline mutation and has been suggested that this may be due to somatic mutations.

Recommended reference: Adeleke S, et al. Microsatellite instability testing in colorectal patients with Lynch syndrome: lessons learned from a case report and how to avoid such pitfalls. Per Med. 2022;19(4):277-286.

A: Thank you for this helpful suggestion. We have revised the Introduction to acknowledge that biological and methodological heterogeneity in MSI/dMMR assessment may complicate the interpretation and comparison of immune-related findings across studies. This revision is supported by the recommended reference and appears on Lines 52–55 of the revised manuscript.

R: “4.4. Prognostic Impact of CD47 Expression on Survival and Recurrence”, Page 6 of 17, Lines 206-208:

“Oh et al. identified a correlation between elevated CD47 expression and reduced OS in univariate analysis; however, this association lacked significance after controlling for age, sex, tumor size, and TNM staging in multivariate Cox regression [21].”.

The authors should include a comment regarding the population of elderly patients. Please note that there is no consensus that age independently affects survival outcomes. The prognosis in older individuals may be influenced by confounding factors such as stage at presentation, tumor site, preexisting comorbidities, and the type of treatment received.

Recommended reference: Osseis M, et al. Surgery for T4 Colorectal Cancer in Older Patients: Determinants of Outcomes. J Pers Med. 2022;12(9):1534.

A: Thank you for this comment. In this section, our intention was to accurately summarize the statistical findings reported by Oh et al., particularly the loss of significance of CD47 expression in multivariate analysis. A detailed discussion of the independent prognostic role of age itself was beyond the scope of this review and therefore was not expanded further in this context.

R: “5. Discussion”, Page 9 of 17, Lines 278-280:

“Although an increasing amount of evidence indicates that CD47 is a promising biomarker and immunotherapy target, there are still some issues that need to be addressed in its clinical translation.”.

At that stage, the authors should note that while tumor-agnostic biomarkers hold great promise for personalized CRC therapy, their clinical implementation faces key challenges. Lack of standardization in molecular testing—stemming from differences in gene panels, sequencing methods, and bioinformatics pipelines—leads to inconsistencies in biomarker detection and interpretation. Additionally, tumor heterogeneity and adaptive resistance, such as bypass signaling in BRAF-mutant CRC, complicate the development of durable treatment strategies. Finally, high testing costs and unequal access to genomic profiling limit the real-world applicability and generalizability of agnostic biomarkers across diverse patient populations.

Recommended reference: Kyrochristou I, et al. Agnostic Biomarkers and Molecular Signatures in Colorectal Cancer-Guiding Chemotherapy and Predicting Response. Biomedicines. 2025;13(8):2038.

A: Thank you for this valuable perspective. We have expanded the discussion to briefly place CD47 within the broader context of challenges faced during the clinical implementation of emerging biomarkers in colorectal cancer, including issues related to testing standardization, tumor heterogeneity, and real-world applicability. This revision is supported by the recommended reference and appears on Lines 332–335 of the revised manuscript.

Reviewer 3 Report

Comments and Suggestions for Authors

The manuscript titled “Immunohistochemical Expression of CD47 in Colorectal Cancer: A Review of Its Clinicopathological Significance” presents a comprehensive overview of CD47 biology, its mechanistic role in immune evasion, and its clinicopathological associations in colorectal cancer. The topic is timely and relevant given the increasing interest in innate immune checkpoints and their therapeutic potential. The authors summarize findings from both immunohistochemical and transcriptomic studies and highlight correlations with tumor stage, molecular subtypes, and the tumor microenvironment. Overall, the review is well-structured and informative; however, several sections would benefit from clearer synthesis, greater critical evaluation of methodological heterogeneity, and improved interpretation of conflicting results. The detailed line-specific comments below are offered to help strengthen clarity, coherence, and scientific rigor.

(Lines 19–23) The statement about "bringing together findings" is broad; consider clarifying that existing studies show substantial heterogeneity in CD47 evaluation.

(Lines 21–23) Add one sentence noting that conflicting IHC positivity rates make interpretation challenging.

(Lines 28–31) Mentioning positivity rates (16–91%) without acknowledging methodological heterogeneity may mislead readers; add a qualifier.

(Line 33) “Co-expression with immune checkpoints and oncogenic pathways” is important; specify one example (e.g., PD-1/PD-L1 or TGF-β).

(Line 37) Strengthen the conclusion by briefly naming one standardization need (antibody clone selection, scoring cut-off).

(Lines 47–53) This section shifts into general immune suppression; streamline to keep focus on CD47 biology in CRC.

(Line 56) When introducing CD47, consider mentioning inconsistencies in CRC-based CD47 studies to justify the review.

(Lines 59–61) The rationale for conducting this review is vague; add what specific knowledge gap you are addressing.

(Lines 84–101) Mechanistic detail is dense; consider adding a brief summary to improve readability for non-specialists.

(Line 93) “Tumor stemness” is mentioned—add CRC-specific evidence or clarify that evidence is extrapolated from other cancers.

(Lines 110–116) Expand implications for CRC (CD47-mediated suppression of antigen presentation affecting MSI-high tumors).

(Line 128) Provide effect sizes (odds ratios or mean differences) where available to contextualize statistical significance.

13. (Lines 158–163) Transcriptomic results are presented descriptively; add a comment on dataset comparability (TCGA vs GEO).

14. (Lines 175–184) Acknowledge sample size disparities across cohorts that may partially explain inconsistent findings.

15. (Lines 208–214) Tian et al. show improved PFI with high CD47—this contradicts most other data; add interpretation for this discrepancy.

16. (Line 217) When stating “independent adverse prognostic factor,” provide HR and CI values to show strength of association.

17. (Lines 249–250) The connection between CMS4 angiogenesis and CD47 could be elaborated to explain biological relevance.

18. (Line 253) CMS1 recurrence and immune checkpoint persistence are mentioned—add how CD47 mechanistically contributes.

19. (Lines 268–273) CD47 correlates with M1 markers but also poor outcomes; add a sentence explaining this paradox (e.g., exhausted M1-like states).

20. (Lines 270–272) Increased CD163+ macrophages suggest immunosuppressive TAM2 enrichment—clarify this implication.

21. (Lines 281–289) Add specific examples showing how different IHC scoring cutoffs (H-score ≥50 vs intensity × area) lead to the wide variability in positivity rates.

22 (Lines 292–299) The point about right- vs left-sided CRC is important but needs supporting citations or explanation of the biological basis.

23. (Lines 308–311) Strengthen the discussion by proposing specific prospective study designs to test causality between CD47 levels and outcomes.

24. (Lines 317–323) Expand how macrophage polarization differences across tumors may alter the prognostic meaning of CD47 expression.

25. (Lines 327–330) The relationship between CD47–SIRPα and PD-1/PD-L1 pathways is highlighted; suggest adding emerging preclinical combination-therapy evidence.

26. (Lines 343–348) Provide at least one example of ongoing clinical trials testing CD47-targeting agents to reinforce translational relevance.

27. (Line 346) When mentioning MSS-type CRC, clarify why CD47 targeting may be useful in checkpoint inhibitor–resistant tumors.

Author Response

REVIEWER 3:

R: The manuscript titled “Immunohistochemical Expression of CD47 in Colorectal Cancer: A Review of Its Clinicopathological Significance” presents a comprehensive overview of CD47 biology, its mechanistic role in immune evasion, and its clinicopathological associations in colorectal cancer. The topic is timely and relevant given the increasing interest in innate immune checkpoints and their therapeutic potential. The authors summarize findings from both immunohistochemical and transcriptomic studies and highlight correlations with tumor stage, molecular subtypes, and the tumor microenvironment. Overall, the review is well-structured and informative; however, several sections would benefit from clearer synthesis, greater critical evaluation of methodological heterogeneity, and improved interpretation of conflicting results. The detailed line-specific comments below are offered to help strengthen clarity, coherence, and scientific rigor.

A: We sincerely thank the reviewer for the thoughtful and constructive comments. In response, we have carefully revised the manuscript to address all points raised, including clarifications of the rationale, refinement of the discussion, and minor textual revisions. All changes are highlighted in the revised manuscript, and a detailed point-by-point response is provided below.

R: (Lines 19–23) The statement about "bringing together findings" is broad; consider clarifying that existing studies show substantial heterogeneity in CD47 evaluation.

A: Thank you for the suggestion. We respectfully note that the Simple Summary is intended for non-specialist readers and is designed to provide a broad, accessible overview rather than methodological details. The discussion of heterogeneity has been addressed thoroughly in the main text; therefore, we have kept the Simple Summary unchanged.

R: (Lines 21–23) Add one sentence noting that conflicting IHC positivity rates make interpretation challenging.

A: Thank you for this suggestion. We agree that acknowledging the inconsistency of reported CD47 expression levels can improve the clarity of the Simple Summary. Therefore, we added a brief sentence noting that previous studies have shown inconsistent expression results. This revision appears in the Simple Summary on Lines 19–20.

R: (Lines 28–31) Mentioning positivity rates (16–91%) without acknowledging methodological heterogeneity may mislead readers; add a qualifier.

A: Thank you for the comment. We would like to note that the current sentence already includes the requested qualifier. The Abstract states that the wide positivity range (16–91%) occurs “due to methodological heterogeneity,” which addresses this point. Therefore, no further changes were made.
This text appears in Lines 32-34 of the revised manuscript.

R: (Line 33) “Co-expression with immune checkpoints and oncogenic pathways” is important; specify one example (e.g., PD-1/PD-L1 or TGF-β).

A: Thank you for the suggestion. Since the Abstract is intended to remain concise, we prefer not to introduce pathway-level examples in this section. These examples are already described in detail in the main text. Therefore, we have left the Abstract unchanged.

R: (Line 37) Strengthen the conclusion by briefly naming one standardization need (antibody clone selection, scoring cut-off).

A: Thank you for this helpful suggestion. We have added a brief specification of key standardization needs to the conclusion of the Abstract. The revised sentence appears on Lines 40–41.

R: (Lines 47–53) This section shifts into general immune suppression; streamline to keep focus on CD47 biology in CRC.

A: Thank you for this comment. We have streamlined this part of the Introduction by removing detailed descriptions of multiple immunosuppressive cell types and retaining only a concise statement about immune evasion. This allows a clearer transition to CD47-related mechanisms.
The revised text appears in Lines 59–61.

R: (Line 56) When introducing CD47, consider mentioning inconsistencies in CRC-based CD47 studies to justify the review.

A: Thank you for the suggestion. We have added a brief statement acknowledging the variability in reported CD47 expression levels in CRC to better justify the need for this review. The revision appears in Lines 61–62.

R: (Lines 59–61) The rationale for conducting this review is vague; add what specific knowledge gap you are addressing.

A: Thank you for this valuable suggestion. We have added a concise statement specifying the knowledge gap—namely the inconsistent findings and methodological variation among CD47 studies in CRC. This revision clarifies the rationale for conducting the review. The updated text appears on Lines 68–70.

R: (Lines 84–101) Mechanistic detail is dense; consider adding a brief summary to improve readability for non-specialists.

A: Thank you for the helpful suggestion. We have added a concise summary sentence at the end of the mechanistic description to improve readability for non-specialist readers. This revision appears in Lines 112–114.

R: (Line 93) “Tumor stemness” is mentioned—add CRC-specific evidence or clarify that evidence is extrapolated from other cancers.

A: Thank you for the suggestion. In this section, our aim was to summarize the general biological functions of CD47 as reported across different tumor types, rather than to present CRC-specific conclusions. Because “tumor stemness” is discussed here as part of the broader mechanistic background and is not used to draw CRC-specific inferences, we have kept the original wording unchanged.

R: (Lines 110–116) Expand implications for CRC (CD47-mediated suppression of antigen presentation affecting MSI-high tumors).

A: Agree. We have revised the manuscript to address this point. Changes made: A sentence has been added to the "Mechanism of CD47-Mediated Immune Escape" section to explicitly link the general mechanism of CD47-mediated antigen presentation suppression to its specific implication in MSI-high CRC (Lines 129-131).

R: (Line 128) Provide effect sizes (odds ratios or mean differences) where available to contextualize statistical significance.

A: Agree. We thank the reviewer for this suggestion to improve the context of the statistical findings. We have revised the manuscript accordingly.

Changes made: Although the original article presents the comparative data graphically (Oh et al., Fig. 1C) without listing exact numerical values in the text, we have added a qualitative description of the effect size to the sentence in question. This provides readers with an immediate understanding of the magnitude of the difference.

Location in the revised manuscript: lines 143-146.

R: (Lines 158–163) Transcriptomic results are presented descriptively; add a comment on dataset comparability (TCGA vs GEO).

A: Agree. We thank the reviewer for this valuable suggestion to strengthen the methodological rigor of our argument. We have revised the manuscript to explicitly address the comparability and collective strength of the transcriptomic evidence.

Changes made: In the paragraph spanning lines 158-163, we have inserted a sentence that comments on the dataset comparability. The addition highlights that the consistent finding of CD47 upregulation across heterogeneous genomic platforms (TCGA and GEO) enhances the reliability of this transcriptional signature, making it a more robust candidate for the clinical applications subsequently discussed. Location in the revised manuscript: lines 189-191.

R: (Lines 175–184) Acknowledge sample size disparities across cohorts that may partially explain inconsistent findings.

A: Agree. We thank the reviewer for raising this critical methodological point regarding sample size disparities. We agree that this is a key factor contributing to the heterogeneity in literature.

Changes made: Rather than inserting a localized comment that might disrupt the narrative flow of the results section, we have incorporated this important consideration into the broader methodological discussion in Discussion. Here, we systematically address how variability in cohort size, alongside differences in IHC protocols and scoring systems, may collectively explain the inconsistent findings across studies. This provides a more integrated and impactful analysis for the reader. Location in the revised manuscript: lines 349-353.

R: (Lines 208–214) Tian et al. show improved PFI with high CD47—this contradicts most other data; add interpretation for this discrepancy.

A: Agree. We thank the reviewer for highlighting this key point. We have revised the section to provide a clearer and more direct interpretation of the discrepancy, drawing explicitly from the explanation offered in the original study by Tian et al.

Changes made: In Section 4.4, we have rephrased the description of Tian et al.’s findings. The revision now: Explicitly frames the result as “paradoxical” given the concurrent association with advanced stage. Directly states that the original authors investigated and noted this discrepancy. Clearly links their observed correlation with M1 macrophages to their proposed resolution—that the immune context can modulate CD47’s prognostic impact. This change ensures that the reader immediately understands why this study appears to contradict others and what the authors’ own data-driven hypothesis is. Location in the revised manuscript: lines 243-251.

R: (Line 217) When stating “independent adverse prognostic factor,” provide HR and CI values to show strength of association.

A: Agree and corrected. We sincerely thank the reviewer for this critical observation. Upon carefully re-examining the source study (Kim et al., 2021) to provide the requested hazard ratios, we identified an inaccuracy in our original summary. We have corrected the manuscript to reflect the precise findings of the multivariate analysis.

Changes made: We have replaced the previous sentence in Section 4.4 with a precise, data-driven summary that distinguishes between univariate and multivariate results, directly citing the hazard ratios and confidence intervals from the original publication (Table 3). Location in the revised manuscript: lines 252-257.

R: (Lines 249–250) The connection between CMS4 angiogenesis and CD47 could be elaborated to explain biological relevance.

A: Thank you for this helpful suggestion. We have revised this section to further elaborate the biological relevance of CD47 in the CMS4 subtype by clarifying its role as an innate immune checkpoint contributing to immune exclusion within the matrix- and angiogenesis-enriched CMS4 tumor microenvironment. This revision appears on Lines 293–296 of the revised manuscript.

R: (Line 253) CMS1 recurrence and immune checkpoint persistence are mentioned—add how CD47 mechanistically contributes.

A: Agree. We thank the reviewer for this suggestion to strengthen the mechanistic link at this point in the narrative.

Changes made: We have expanded the sentence to explicitly state how CD47 mechanistically contributes to the observed clinical pattern. The addition concisely summarizes its role as an innate checkpoint that suppresses phagocytosis and antigen presentation and proposes how this function may synergize with adaptive checkpoints to drive post-recurrence immune evasion.

Location in the revised manuscript: Line 298-302, at the end of the sentence discussing CMS1 recurrence and checkpoint persistence.

R: (Lines 268–273) CD47 correlates with M1 markers but also poor outcomes; add a sentence explaining this paradox (e.g., exhausted M1-like states).

A: Agree. We have added a concise explanation for the noted paradox.

Changes made: We added a sentence proposing that the co-occurrence of M1 markers and immunosuppression may indicate the presence of dysfunctional, exhausted macrophages, potentially driven by persistent CD47-SIRPα signaling.

Location: Line 326-328.

R: (Lines 270–272) Increased CD163+ macrophages suggest immunosuppressive TAM2 enrichment—clarify this implication.

A: Agree. We thank the reviewer for this suggestion to strengthen the interpretation. We have clarified the implication of increased CD163+ macrophages.

Changes made: We have inserted a sentence between the report of the findings and its broader interpretation. This new sentence explicitly states that CD163 is a marker for immunosuppressive (M2-like) TAMs and proposes that CD47 may play an active role in fostering this cell population, thereby shaping the immunosuppressive TME.

Location in the revised manuscript: Lines 321-323.

R: (Lines 281–289) Add specific examples showing how different IHC scoring cutoffs (H-score ≥50 vs intensity × area) lead to the wide variability in positivity rates.

A: Agree. We have added a concise illustrative example as suggested.

Changes made: We inserted a brief hypothetical example comparing two scoring approaches (H-score vs. intensity-focused) to show how the same sample could be classified differently, directly explaining part of the variability in positivity rates. Location: Line 343-348.

R: (Lines 292–299) The point about right- vs left-sided CRC is important but needs supporting citations or explanation of the biological basis.

A: We have added a brief biological rationale.
Change made: Added a sentence linking the sidedness difference to embryological origin and its impact on the immune system.
Location: Line 362-364.

R: (Lines 308–311) Strengthen the discussion by proposing specific prospective study designs to test causality between CD47 levels and outcomes.

A: Thank you for this comment. We have strengthened the Discussion by adding a concise statement highlighting the need for prospective studies with standardized CD47 assessment to clarify causal relationships. This revision appears in the Discussion on Lines 378–380.

R: (Lines 317–323) Expand how macrophage polarization differences across tumors may alter the prognostic meaning of CD47 expression.

A: Agree. We thank the reviewer for prompting a more detailed exposition of this key concept. We have expanded the discussion to explicitly outline how macrophage polarization can alter the prognostic interpretation of CD47.

Changes made: In the paragraph discussing the environmental dependency of CD47, we have added a dedicated explanatory passage. This addition provides a specific two-scenario model: (1) in an M2-dominant TME, high CD47 worsens immune suppression and prognosis; (2) in an M1-inflamed TME, high CD47 may be a compensatory marker, and its prognostic meaning is confounded by the dominant anti-tumor response. This directly addresses how polarization differences can change CD47’s prognostic meaning.

Location in the revised manuscript: Line 392-400.

R: (Lines 327–330) The relationship between CD47–SIRPα and PD-1/PD-L1 pathways is highlighted; suggest adding emerging preclinical combination-therapy evidence.

A: We thank the reviewer for this suggestion. As requested, we have now added a discussion of recent preclinical evidence supporting the dual blockade strategy. We cite a study on a CD47/PD-L1 bispecific antibody (6MW3211) that shows potent synergy in vivo and a favorable safety profile in primates, which validates the translational potential of this approach. We have connected this to our work by stating: “In this context, our discovery of SMC18… provides a complementary therapeutic approach.” This addition appears in the Discussion section (Lines 425–435) of the revised manuscript

R: (Lines 343–348) Provide at least one example of ongoing clinical trials testing CD47-targeting agents to reinforce translational relevance.

A: We thank the reviewer for this suggestion. We have added an example of an ongoing clinical trial to reinforce translational relevance. We now cite the phase I/II study NCT04588324, which combines a CD47 antibody with the small molecule immunomodulator SHR2150 and chemotherapy in advanced solid tumors. This example validates the clinical exploration of CD47-targeting combinations and supports the development rationale for our dual-targeting compound SMC18. The addition is in the Discussion section (Lines 435-441).

R: (Line 346) When mentioning MSS-type CRC, clarify why CD47 targeting may be useful in checkpoint inhibitor–resistant tumors.

A: Thank you for this comment. To avoid overinterpretation and to maintain consistency with the evidence discussed in the main text, we have removed the reference to MSS-type CRC in the Conclusion and retained CMS4 CRC, which is directly supported by the reviewed data. The revised sentence appears in Lines 471.

Round 2

Reviewer 3 Report

Comments and Suggestions for Authors

The manuscript is now significantly improved, and I believe it meets the standards for publication. I have no further concerns, and I recommend the paper for acceptance in its current form.